# 3D-Cultured Adipose-Derived Stem Cell Spheres Using Calcium-Alginate Scaffolds for Osteoarthritis Treatment in a Mono-Iodoacetate-Induced Rat Model

**DOI:** 10.3390/ijms24087062

**Published:** 2023-04-11

**Authors:** Yu-Ying Lin, Che-Yung Kuan, Chia-Tien Chang, Ming-Hsi Chuang, Wan-Sin Syu, Kai-Ling Zhang, Chia-Hsin Lee, Po-Cheng Lin, Guo-Chung Dong, Feng-Huei Lin

**Affiliations:** 1Ph.D. Program in Tissue Engineering and Regenerative Medicine, National Chung Hsing University, Taichung 40227, Taiwan; butter97132195@gmail.com; 2Institute of Biomedical Engineering and Nanomedicine, National Health Research Institutes, Miaoli County 35053, Taiwan; cheyungkuan@ntu.edu.tw (C.-Y.K.); aaiaaiaai0813@gmail.com (C.-T.C.); 3Institute of Biomedical Engineering, College of Medicine and College of Engineering, National Taiwan University, Taipei 10087, Taiwan; 4College of Management, Chung Hwa University, Hsinchu 30012, Taiwan; mercy@gwoxi.com; 5Gwo Xi Stem Cell Applied Technology, Hsinchu 30261, Taiwan; virllyck@gmail.com (W.-S.S.); kailing.zhang@gwoxi.com (K.-L.Z.); jessie.lee@gwoxi.com (C.-H.L.); genecold@gwoxi.com (P.-C.L.); 6College of Biological Science and Technology, National Yang Ming Chiao Tung University, Hsinchu 30010, Taiwan

**Keywords:** osteoarthritis, human adipose-derived stem cells, cell spheres, alginate, monosodium iodoacetate

## Abstract

Osteoarthritis (OA) is a degenerative disease that causes pain, cartilage deformation, and joint inflammation. Mesenchymal stem cells (MSCs) are potential therapeutic agents for OA treatment. However, the 2D culture of MSCs could potentially affect their characteristics and functionality. In this study, calcium-alginate (Ca-Ag) scaffolds were prepared for human adipose-derived stem cell (hADSC) proliferation with a homemade functionally closed process bioreactor system; the feasibility of cultured hADSC spheres in heterologous stem cell therapy for OA treatment was then evaluated. hADSC spheres were collected from Ca-Ag scaffolds by removing calcium ions via ethylenediaminetetraacetic acid (EDTA) chelation. In this study, 2D-cultured individual hADSCs or hADSC spheres were evaluated for treatment efficacy in a monosodium iodoacetate (MIA)-induced OA rat model. The results of gait analysis and histological sectioning showed that hADSC spheres were more effective at relieving arthritis degeneration. The results of serological and blood element analyses of hADSC-treated rats indicated that the hADSC spheres were a safe treatment in vivo. This study demonstrates that hADSC spheres are a promising treatment for OA and can be applied to other stem cell therapies or regenerative medical treatments.

## 1. Introduction

Osteoarthritis (OA) is the most common musculoskeletal disease and was assumed to be the fourth leading cause of disability worldwide in 2020 [1,2]. OA can cause a variety of discomforts, including pain, stiffness, and dysfunction, which can lead to chronic disability and significant financial burden, especially in people over the age of 65 [3]. OA affects the knee joint the most, accounting for nearly 85% of the global burden of OA, followed by the joints of the hands and hips [2]. In addition, owing to the increase in the global average life expectancy, the prevalence of OA is also increasing, which has a significant societal impact [4].

Generally, non-pharmacological methods are regarded as the primary means of treating knee OA, including education, self-management, regular exercise, and weight control, especially in obese people [2,5]. However, when non-pharmacological methods cannot relieve pain and reduce disability in patients, pharmacological methods are recommended to treat OA, such as paracetamol and non-steroidal anti-inflammatory drugs (NSAIDs) [5]. However, long-term NSAID use may cause kidney damage, myocardial infarction, gastric irritation, and gastrointestinal bleeding [6,7].

Intraarticular (IA) injection is an alternative strategy for knee OA therapy. Corticosteroids (CS), hyaluronic acid (HA), and platelet-rich plasma (PRP) have been used for the management of knee OA. CS injection was shown to reduce pain within 1 month post-injection; HA injections required almost 2–3 months to relieve pain, while PRP injections significantly reduced pain in knee OA patients within 3 months [8,9]. However, these types of IA injections only show analgesic and lubrication effects and do not lead to recovery of cartilage cells in the OA area. Therefore, the pain relief associated with IA injection diminishes weeks after injection [7,10,11,12].

Cartilage is a soft connective tissue composed of extracellular matrix (ECM) synthesized by resident chondrocytes; therefore, maintaining the phenotype of healthy chondrocytes is important for the quality of ECM maintenance [13]. Nevertheless, self-repair of articular cartilage (AC) is difficult because of its low regenerative capacity caused by the lack of blood supply, low cellularity, and a limited number of progenitor cells. Moreover, autograft cartilage cells also have limitations, such as the small number of cells available and low chondrogenic ability [14]. Stem cell treatment is another approach for the treatment of knee OA. Among the various cell therapies, adipose-derived stem cell (ADSC) therapy is considered promising. ADSCs can differentiate into different mesenchymal cell types, such as bone, cartilage, and adipocytes [15]. In addition, ADSCs are an abundant source of multi-potent adult stem cells that are easily isolated from subcutaneous fat tissue using minimally invasive surgery [16]. Therefore, ADSCs are potential candidates for knee OA treatment.

Two-dimensional (2D) culture conditions vary widely for each cell type. In 2D culture, the proliferated cells are grown on the surface of culture flasks, plates, or dishes and the cell number is limited by the surface area of the culture container. In general, using 2D culture, it is not possible to reach the cell number required for clinical usage. In addition, the 2D culture approach has been demonstrated to present critical limitations, resulting in low differentiation efficiency [17]. To overcome these limitations, a three-dimensional (3D) cell culture method was developed to improve the cells and physiological equivalence of in vitro experiments [18]. The use of various dynamic in vitro 3D tissue culture systems to mimic the native microenvironment of target tissues has been explored for decades [19]. Scaffold-based culture technologies offer physical support to mechanical structures in ECM-like matrices, on which isolated cells can migrate, proliferate, and aggregate [20]. Moreover, the key functions of these scaffolds are to encourage cell-scaffold interactions; promote cell adhesion; permit adequate transport of gases and nutrients to ensure cell survival, proliferation, and differentiation under a negligible amount of inflammation or toxicity; and control the structure and function of cells [21]. Natural polymer scaffolds usually demonstrate a lack of immune response and improved cell interactions compared to synthetic polymers [22]. 

Alginate is a natural polysaccharide composed of β-D-mannuronate and α-L-glucuronate that is widely used in tissue engineering [23,24,25]. Alginate can form hydrogels via non-covalent cross-linking with divalent cations [17]. Alginate scaffolds can also exhibit high porosity, allowing the transportation of gases and nutrients via a simple freeze-drying method. Moreover, alginate hydrogels have been demonstrated to support chondrocyte growth and chondrogenesis of mesenchymal stem cells [26,27]. 

In this study, alginate was cross-linked with calcium ions to form a hydrogel and a porous scaffold was obtained by freeze-drying. The harvested human adipose-derived stem cells (hADSCs) were seeded onto Ca-Ag scaffolds and proliferated in a homemade functionally closed process bioreactor system. The cells could simultaneously form hADSC spheres on the Ca-Ag scaffolds and the hADSC spheres could be collected by removal of calcium ions via ethylenediaminetetraacetic acid (EDTA) chelation. In this study, individual 2D-cultured hADSCs or hADSC spheres were evaluated for knee OA treatment. The monosodium iodoacetate (MIA)-induced OA model was used to prove the concept and evaluate the efficacy of the developed scaffold in vivo via hematoxylin and eosin (HE) and toluidine blue staining. Gait analysis was conducted to evaluate the walking performance of the hADSC-treated MIA-induced OA rats. The OA score was evaluated using the Osteoarthritis Research Society International (OARSI) osteoarthritis cartilage histopathology assessment system. The overall process is schemed as shown in Figure 1. We hypothesized that hADSC spheres would have more potential to treat knee OA than 2D-cultured hADSCs. 

## 2. Results

### 2.1. Characterization of Ca-Ag Scaffolds

The Ca-Ag scaffold were prepared by freeze-drying technique. The scaffolds had a cylindrical shape with a diameter of 5.87 ± 0.29, height of 6.13 ± 0.34 mm, and dry weight of 12.35 ± 0.54 mg (Figure 2a). The swelling scaffolds were formed after the scaffolds absorbed PBS (Figure 2b). The FT-IR spectrum of Ca-Ag scaffold is shown in Figure 2c. Stretching vibrations of O-H bonds of alginate appeared in the range of 3000–3650 cm^−1^. Stretching vibrations of aliphatic C-H were observed at 2920 cm^−1^. The bands obtained at 1595 and 1470 cm^−1^ were contributed from asymmetric and symmetric vibrations of carboxylate salt ion, respectively. The band of 973 cm^−1^ was contributed from C-O-C group. The Ca-Ag scaffolds were confirmed to be dissolvable in 50 mM EDTA solution for 5 min at 37 °C (Figure 2d,e).

### 2.2. Seeding Efficiency and Cell Proliferative Quantification

To determine seeding efficiency of hADSCs on Ca-Ag scaffolds, CellTiter-Glo^®^ Luminescent Cell Viability Assay was used to estimate the cell number after the hADSCs cultured with the scaffolds for 1 day. Initial cell density was 1 × 10^6^ cells/scaffold and cell seeding efficiency was about 82.67%. The proliferation of hADSCs adhered to the Ca-Ag scaffold was determined after culturing with bioreactor on day 14. The 2.35 fold cell growth from the initial cell density (Appendix A). The hADSC spheres were observed in the Ca-Ag scaffold at day 14 by a SEM (Figure 3).

### 2.3. Gait Analysis

The study was divided into four groups described and abbreviated as follows: (1) the rats only received normal saline injection (Sham); (2) the rats received MIA injection (MIA); (3) the rats received MIA injection and were treated with individual cells (2D); and (4) the rats received MIA injection and were treated with cell spheres (3D).

To explore the effect of hADSC spheres on walking performance in MIA-induced OA rats, gait analysis was conducted. The results showed that the response times for stance, brake, and propulsion in MIA-induced OA rats were higher than those in the Sham Group. Concurrently, the response times of stance, brake, and propulsion in the 2D and 3D groups decreased compared to those of the MIA Group, while the reaction time of the 3D Group was much lower than that of the 2D Group; these results indicate that the hADSC spheres could improve the walking performance of MIA-induced OA rats (Figure 4). 

### 2.4. H&E Stain

Analysis of rat knee joint histological section showed that Sham Group had smooth surface and regular arrangement of chondrocytes in articular cartilage after 4 weeks postoperatively in this animal study. H&E staining of the femorotibial section showed a reduction in the number of chondrocytes and appearance of empty lacunae after MIA treatment, indicating that the chondrocytes were damaged. In the sections of 2D and 3D groups, chondrocytes were partially protected in MIA-induced OA rats (Figure 5). The empty lacunae and chondrocyte of H&E section were illustrated in Appendix A.

H&E staining results at 8 weeks post-operatively indicated the Sham Group maintained the normal cartilage phenotype. However, the number of chondrocytes reducing was observed in the superficial and transition zones of the articular cartilage in the MIA group. The OA rats treated with individual hADSCs or hADSC spheres showed improving phenotype compared to the MIA Group. In addition, the 3D Group kept the cartilage phenotype (Figure 5). Overall, these results verified that hADSC spheres had a better effect in relieving arthritis degeneration compared with the 2D Group.

### 2.5. Toluidine Blue Stain and OA Score

The results of toluidine blue staining 4 weeks after surgery are shown in Figure 6. The length from the cartilage surface to the tidemark was measured in the Sham, MIA, 2D, and 3D groups as 131.81, 64.47, 51.32, and 153.95 μm, respectively. At 8 weeks postoperatively, the measured lengths from the cartilage surface to tidemark were 192.10, 73.69, 63.16, and 184.87 μm in the Sham, MIA, 2D, and 3D groups, respectively (Figure 4). The toluidine blue-positive cartilage was only present in the Sham and 3D groups, which indicates that the cartilage could restore the ability to synthesize ECM. 

The sections were further evaluated for OA scores. The scores at 4 weeks after surgery were 0 ± 0, 14 ± 2.31, 11.5 ± 6.40, and 0.75 ± 0.96 in the Sham, MIA, 2D, and 3D groups, respectively (Figure 7a). At 8 weeks postoperatively, the scores of the Sham, MIA, 2D, and 3D groups were 0 ± 0, 16 ± 1.63, 12.5 ± 5.74, and 0.5 ± 0.57, respectively (Figure 7b).

From these results, we conclude that hADSC spheres are a better choice for OA treatment than 2D-cultured cells.

### 2.6. Serological and Blood Elements Analysis

To evaluate the safety of the hADSC treatment, serological and blood element analyses were performed. Blood samples were collected from the tail veins of rats at 4 (Table 1) and 8 weeks (Table 2). The results indicated that the aspartate aminotransferase (GOT) increased in the MIA and 2D groups at 4 weeks. Moreover, the results showed no significant difference between the 3D and Sham groups. These results indicate that hADSCs spheres injected into the joint cavity of rats did not trigger tissue inflammation and showed no liver or kidney toxicity.

## 3. Discussion

Stem cells have been widely used in tissue regeneration owing to their multi-differentiation and self-renewal abilities [28]. Mesenchymal stem cells (MSCs) are multi-potent stem cells capable of differentiating into diverse mesodermal lineages of osteoblasts, chondrocytes, and adipocytes [29]. ADSCs belong to the MSC family and are derived from adipose tissues. Unlike bone marrow mesenchymal stem cells (BMSCs), ADSCs can be easily isolated through liposuction, which is minimally invasive and less painful. Moreover, compared with BMSCs, ADSCs can be harvested at a level of 100–1000 times the number of cells under the same extraction volume and can be a potential candidate for stem cell therapy [30,31]. In this study, we isolated ADSCs from human adipose tissue and used OA treatments for rats for the following reasons: (1) adipose tissue is an abundant source of hADSCs; (2) it has been reported that hADSCs do not express MHC class II, i.e., hADSCs do not trigger a xenogenic immunologic reaction [32]; and (3) the final goal of our OA treatment strategy is to use human autologous hADSCs to treat OA.

Animal models of OA can be divided into naturally occurring OA, surgically induced OA, and chemically induced OA. Spontaneous OA models do not require any interventions to induce this condition. However, the limitation is inherent variability, which incurs greater numbers and costs [33]. A variety of surgically induced models have been reported and the commonly used models for rodents include meniscectomy, destabilization of the medial meniscus (DMM), meniscal tear, and anterior cruciate ligament (ACL) or posterior cruciate ligament transection [34]. However, some surgeries require well-trained operators to avoid experimental variation. Chemically induced models involve injection of toxic or inflammatory compounds directly into the knee joint. This is a less invasive and more easily performed method for inducing OA. The severity of the induced OA can also be evaluated and optimized through dosage control. Moreover, the model only required a single injection for OA induction; the commonly used agents included immunotoxins, collagenase, papain, and MIA. MIA is the most commonly used compound in OA studies and is an inhibitor of glyceraldehyde-3-phosphate dehydrogenase of the Krebs cycle, leading to the death of chondrocytes. This model generates a reproducible, robust, and rapid pain-like phenotype that can be graded by altering MIA dosage [35,36].

Cell-based therapy is increasingly popular and has attracted much attention for the treatment of incurable diseases. However, a major limitation of cell-based therapy is the need for large cell numbers. It is estimated that up to 10^9^ cells are required for a single treatment. Currently, it is difficult to harvest sufficient cell numbers for medical use using traditional cell culture methods [37,38]. For example, the maximum yield of cells harvested from a 75T flask is approximately 10^7^ cells. Hence, for cell transplantation treatment, the operator needs to prepare more than 100 T75 flasks to reach 100 million cells, which could cause batch variability, inefficiency, and low quality of transplantable cells in 2D culture. In addition, these processes also have the risk of exposure to contamination. Therefore, the bioreactor system plays a key role in cell expansion in cell-based therapy for clinical translation. 

The defined bioreactor was designed to mimic the physiological conditions to provide a good environment for the cells to adhere, survive, and proliferate on the scaffolds [39]. In our previous study, a closed bioreactor system was designed to culture osteoblast cell clusters [17]. In this study, hADSCs were seeded onto the Ca-Ag scaffold and cultured in our homemade functionally closed process bioreactor system. The hADSC spheres were formed on the Ca-Ag scaffolds and grown in the bioreactor under a constant medium flow supplying nutrition to maintain cell survival and proliferation in the scaffolds. The hADSC spheres were collected and compared with 2D-cultured hADSCs for OA treatment. These results verified that hADSC spheres had a better effect in relieving arthritis degeneration compared with the 2D-cultured hADSCs. However, the cell proliferation rate was a limitation in this study. There was only 2.35 times cell growth from the initial cell density obtained after culturing with bioreactor for 14 days. Through modification of the surface area of scaffold, the cell seeding method and nutrition supply of bioreactor could improve the proliferation rate in this bioreactor system. 

ADSC spheroids have been suggested to improve cell biological properties, including increasing cell viability and proliferative capacity, stabilizing morphology, and improving metabolic functions, compared to 2D cultures [40]. In addition, it has been reported that 3D cultures can enhance differentiation markers and anti-inflammatory cytokine gene expression and stimulate ECM secretion [14]. In this study, the walking performance of 3D hADSC-treated MIA-induced OA rats were significantly improved compared to that of rats administered 2D-cultured hADSCs (Figure 4). The femorotibial section showed that the 3D hADSC sphere-treated groups had the least chondrocyte damage and greatest ability to synthesize ECM (Figure 5 and Figure 6). The OA score was also significantly reduced, indicating recovery of OA rats (Figure 7). The results of the animal study suggested that hADSC sphere treatment had a better effect than individual cells in relieving arthritis degeneration in MIA-induced rats; it also exhibited a chondrocyte protection effect. Moreover, according to the serological and blood element analysis of hADSC-treated OA rats, no inflammation indicators for liver and kidney toxicity were observed, indicating that the hADSC spheres were a safe treatment in vivo (Table 1 and Table 2). In summary, these results suggest that hADSC spheres are a promising treatment for OA and can be applied to stem cell therapies or other applications in regenerative medicine. 

## 4. Materials and Methods

### 4.1. Materials

Sodium alginate (100–300 cP, 2% (25 °C)), calcium chloride, EDTA, formaldehyde, monosodium iodoacetate (MIA), sodium chloride, xylene, hematoxylin, eosin Y, and toluidine blue were purchased from Sigma-Aldrich (St. Louis, MO, USA). hADSCs was purchased from Invitrogen (Waltham, MA, USA), while CellTiter-Glo^®^ Luminescent Cell Viability Assay kit was purchased from Promega (Madison, WI, USA). Isoflurane was purchased from Panion & BF Biotech (Taipei, Taiwan), while Ethanol was purchased from Echo chemical (Miaoli, Taiwan). Povidone-iodine was purchased from Jen Sheng pharmaceutical Co., Ltd. (Taichung, Taiwan) 

### 4.2. Preparation of Ca-Ag Scaffolds

Ca-Ag scaffolds were prepared using a freeze-drying method, as described previously [17]. In brief, 1.5% (*w/v*) of sodium alginate (A2158, Sigma, St. Louis, MO, USA) was dissolved in de-ionized water at room temperature; the solution was then transferred into a 48-well culture plate with a volume of 1 mL per well. The culture plate was frozen at −20 °C overnight and porous alginate sponges were obtained using a freeze-dryer. The dried alginate sponges were cross-linked with 2% calcium chloride (C1016, Sigma, USA) solution for 1 h at room temperature to prepare Ca-Ag scaffolds. The cross-linked Ca-Ag scaffolds were sterilized with 75% ethanol (48840001041-06-75EC, Echo chemical, Taiwan), dried in a gradient series of ethanol, and stored at room temperature for later use. A FT-IR spectroscopy (Spectrum 100 FT-IR Spectrometer, PerkinElmer, Waltham, MA, USA) with auto-attenuated total reflectance (ATR) accessory was used to detect the functional group of Ca-Ag scaffold. The spectra were recorded wavelength between 4000 to 600 cm^−1^ with a resolution of 8 cm^−1^; the number of scans performed was 4.

### 4.3. Seeding of hADSCs on Ca-Ag Scaffolds

hADSCs (StemPro^®^ Human Adipose-Derived Stem Cells, R7788-115, Invitrogen, USA) were generously provided by Gwo Xi Stem Cell Applied Technology. The hADSCs were suspended in the medium and seeded into scaffolds at a density of 1 × 10^6^ cells/scaffold. The scaffolds with hADSCs were placed in a 24-well culture plate for 24 h at 37 °C under 5% CO_2_ for cell adhesion and transferred to a functionally closed process bioreactor system. The medium was circulated at an initial pump setting of 1 mL/min using a peristaltic pump. The hADSCs spheres were obtained from Ca-Ag scaffolds by dissolving the scaffolds in 50 mM EDTA (324503, Sigma, USA) solution for 5 min at 37 °C.

### 4.4. Seeding Efficiency and Cell Proliferative Quantification

The seeding efficiency of hADSCs on Ca-Ag scaffolds was quantified by CellTiter-Glo^®^ Luminescent Cell Viability Assay (G7570, Promega, USA). The hADSCs were briefly seeded into Ca-Ag scaffolds at a density of 1 × 10^6^ cells/scaffold. The scaffolds with hADSCs were placed in a 24-well culture plate for 24 h at 37 °C under 5% CO_2_ for cell adhesion. The scaffolds were transferred into another well and treated with 50 mM EDTA solution for 5 min at 37 °C to obtain hADSCs spheres. The supernatants were discarded after centrifuged for 5 min at 1200 rpm. Both the hADSCs spheres and cultured medium were reacted with 0.5 mL of CellTiter-Glo^®^ reagent for 30 min in a dark environment. The luminescent intensities were recorded using multilabel plate readers (EnSpire, PerkinElmer, USA). The seeding efficiency was calculated by the following formula: $\mathrm{seeding}\;\mathrm{efficiency}\;\left(\%\right)=100\times I_{hADSCs\;}/\left(I_{hADSCs\;}+I_{cultured\;medium}\right)$, where *I* is the luminescent intensities.

Cell proliferative quantification was performed at the end point. The scaffolds were collected from bioreactor on day 14 and transferred into a 50 mL tube. The scaffolds were then dissolved for obtaining the hADSCs spheres by treating 50 mM EDTA solution for 5 min at 37 °C. The supernatants were discarded after being centrifuged for 5 min at 1200 rpm. The hADSCs spheres were reacted with 0.5 mL of CellTiter-Glo^®^ reagent for 30 min in a dark environment. The luminescent intensities were recorded using multilabel plate readers (EnSpire, PerkinElmer, USA).

### 4.5. SEM Analysis

The morphology of hADSC spheres on Ca-Alginate scaffolds was observed by a SEM (TM-1000, Hitachi, Tokyo, Japan). The sample was fixed with 4% formaldehyde (HT501128, Sigma, USA) for 2 h and dehydrated using a graded ethanol series (30–100%). All the samples were dried by critical point drying method and sputter-coated with platinum. 

### 4.6. Animal Study

#### 4.6.1. Rat OA Model

Male Sprague–Dawley rats (176–200 g) were used in this study. The rats were purchased from BioLASCO, Taiwan, and delivered to the Laboratory Animal Center, National Health Research Institutes, Taiwan, seven days before the experiment began to accommodate the environment. One cage per two rats was utilized throughout the experimental period with a controlled temperature and humidity of 22 °C and 55%, respectively; the period also used a 12-h light-on and -off method. The study protocol was approved by the Institutional Animal Care and Use Committee of the National Health Research Institutes (NHRI-IACUC-107115).

To induce MIA-induced OA, rats were anesthetized with isoflurane (Panion & BF Biotech, Taiwan) and given a single intra-articular injection of 0.3 mg monosodium iodoacetate (MIA, I2512, Sigma, USA) prepared in 50 μL of sterile normal saline (S5866, Sigma, USA) through the infrapatellar ligament of the left knee. The rats were anesthetized with isoflurane and the injected area was sterilized using 75% ethanol and povidone–iodine (F18030, Jen Sheng pharmaceutical Co., Ltd., Taiwan). MIA was injected using a syringe with a 27-gauge needle. After 3 days of MIA treatment, the rats were anesthetized with isoflurane and 50 μL of individual cells (1 × 10^6^), cell spheres (1 × 10^6^), or normal saline was injected into the same MIA-induced location. 

Gait analysis was performed using a Treadscan 4 weeks after cell injection. At the end of the experiment, the rats were sacrificed and blood was collected directly.

#### 4.6.2. Serological and Blood Elements Analysis

For serological analysis, blood was collected in a blood collection tube (450533, Greiner Bio-One, Kremsmünster, Austria). The collection tubes were centrifuged for 10 min at 3500 rpm (5500, Kubota, Osaka, Japan). The supernatant was collected and analyzed for liver function (AST and ALT), alkaline phosphatase (ALP), and renal function (BUN and CRE) using a serology analyzer (DRI-CHEM NX-500 I, Fujifilm, Tokyo, Japan).

For blood element analysis, blood was collected in a purple collection tube containing an EDTA anticoagulant and mixed homogeneously for analysis. The white blood cells (WBCs), red blood cells (RBCs), hemoglobin (HGB), hematocrit ratio (HCT), platelets (PLT), neutrophils (NE), eosinophilic multinuclear (EO), basophil (BA), lymphocytes (LY), and mononuclear spheres (MO) were analyzed using a hematology analyzer (BC-5000 VET, Mindray, Shenzhen, China).

#### 4.6.3. Histological Staining

Knee joint tissue samples were harvested using a sterilized surgical instrument. The surrounding soft tissue samples were carefully trimmed and cleaned with PBS. The samples were placed in a 4% formaldehyde solution (HT501128, Sigma, USA) for fixation and subsequently decalcified in 5% nitric acid for 2 weeks. The tissue was embedded in paraffin in a tissue embedder (Tissue-Tek TEC-6, Sakura Finetek, Torrance, CA, USA) and the paraffin blocks were sectioned (5 mm-thick sections) using a rotary microtome (RM 215, Leica, Nußloch, Germany). 

For H&E staining, the sections were deparaffinized with xylene (534056, Sigma, USA), followed by serial ethanol rehydration. The slides were then placed in a Coplin jar containing a hematoxylin solution (GHS3, Sigma, USA) for 5 min and rinsed with ddH_2_O for 2 min. The sections were stained with 1% eosin Y solution (E4382, Sigma, USA) for 3 min and dehydrated two times with 95% ethanol and two changes of absolute ethanol. The sections were cleaned with xylene for 5 min and placed on the cover slide with mounting media. Images were observed using an optical microscope (Eclipse 80i, Nikon, Tokyo, Japan). 

Toluidine blue staining was used for knee joint evaluation after cell treatment. The sections were briefly deparaffinized using xylene, followed by serial ethanol rehydration. The slides were dipped into a Coplin jar containing 0.04% of toluidine blue solution (T3260, Sigma, USA) for 10 min and rinsed with ddH_2_O for 1 min. The sections were then dried at room temperature and washed with xylene for 5 min. The sections were covered with a cover slide using mounting medium. The images were observed using an optical microscope and the OA score was evaluated by toluidine blue staining following the Osteoarthritis Research Society International (OARSI) osteoarthritis cartilage histopathology assessment system. The OA score index for the combined grade and stage was recommended. The OA score was calculated using the following formula: score = grade × stage, with a range of 0–24 based on the most advanced grade and most extensive stage [41].

### 4.7. Statistic Method

All the results and data were presented with means ± standard deviation. Statistical analyses were performed by one-way ANOVA. The results were considered of significant difference when the *p*-value < 0.05.

## 5. Conclusions

In this study, we successfully developed Ca-Ag scaffolds for human adipose-derived stem cell (hADSC) proliferation and incubated them with a homemade functionally closed process bioreactor system. These results provide an alternative therapeutic strategy for intra-articular knee OA therapy. In addition, the results of the in vivo study indicate that hADSC sphere treatment improved the walking performance and OA score of MIA-induced OA rats. These findings fully support the argument that hADSC heterologous stem cell therapy is safe and effective for OA treatment.

## Figures and Tables

**Figure 1 ijms-24-07062-f001:**
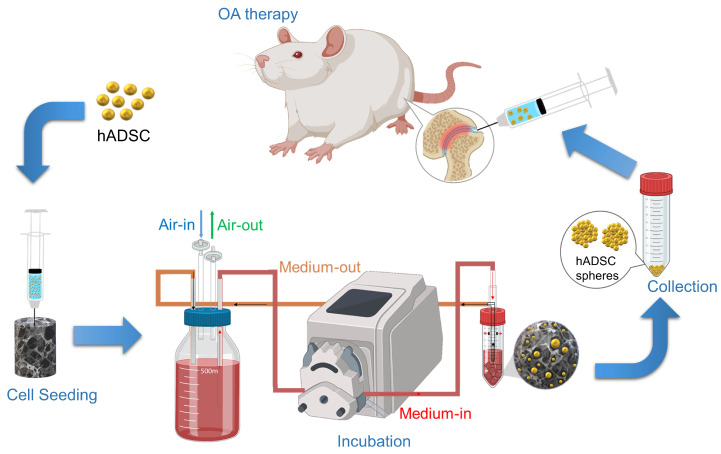
The graph illustrates the concept and strategy for this study. Briefly, the first step was to seed hADSC into Ca-Ag scaffolds and incubate at a functionally-closed process bioreactor system. At the end of incubation, the hADSC spheres were collected by non-enzymatic treatment and centrifugation. The hADSC spheres were injected into the articular cavity for OA therapy.

**Figure 2 ijms-24-07062-f002:**
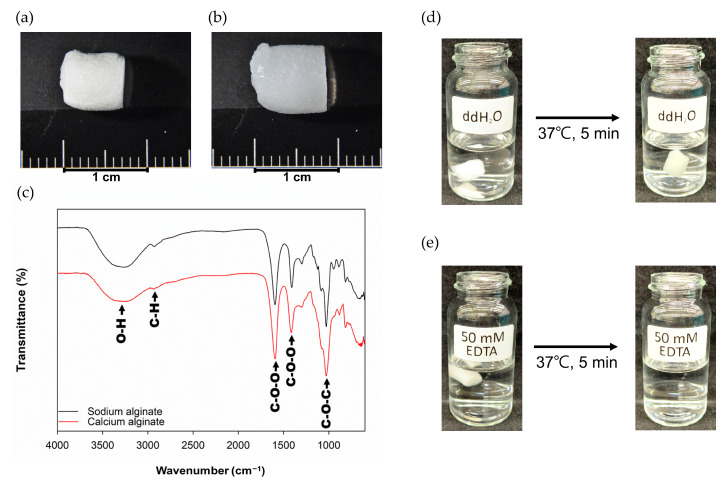
Characterization of Ca-Alginate Scaffolds: (**a**) the Ca-Ag scaffold was cylinder-shaped and the average dimensions were 6.13 mm in height and 5.87 mm in diameter; (**b**) image of Ca-Ag scaffold after immersed in ddH_2_O for 5 min; (**c**) FT-IR spectrum of Ca-Ag scaffold; (**d**) image of Ca-Ag scaffold treated with ddH_2_O for 5 min at 37 °C; (**e**) image of Ca-Ag scaffold treated with 50 mM EDTA for 5 min at 37 °C, at which point the scaffold was fully dissolved.

**Figure 3 ijms-24-07062-f003:**
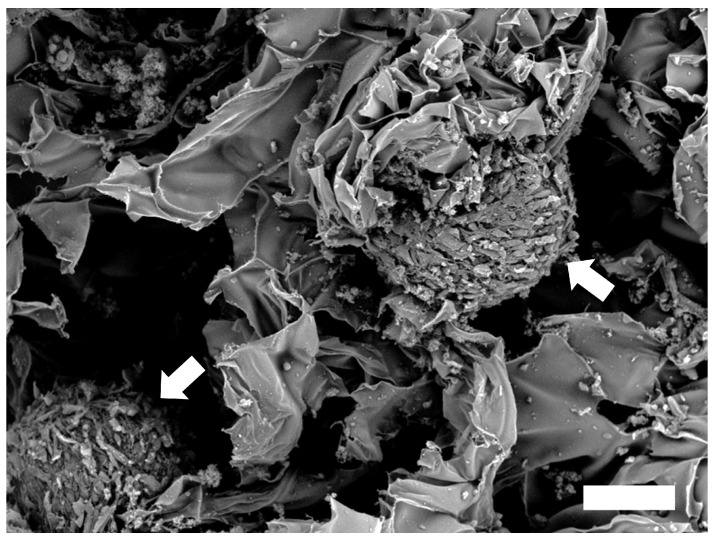
Observation of hADSC spheres (white arrow) formed in the Ca-Ag scaffold at day 14 by scanning electronic microscope (scale bar = 100 μm).

**Figure 4 ijms-24-07062-f004:**
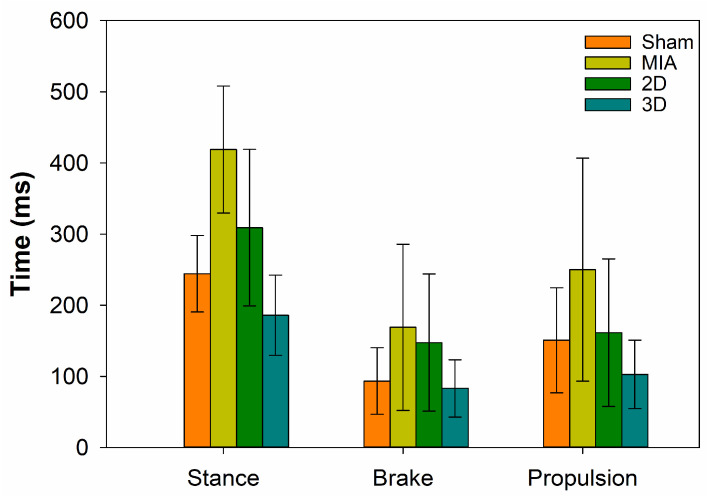
Evaluation of walking performance by gait analysis (n = 4).

**Figure 5 ijms-24-07062-f005:**
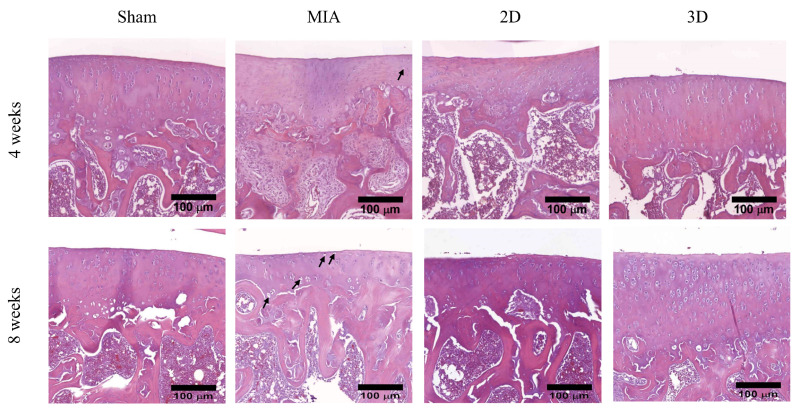
Articular cartilage of rat knee with H & E staining. The empty lacunae were indicated by black arrow (scale bar = 100 μm).

**Figure 6 ijms-24-07062-f006:**
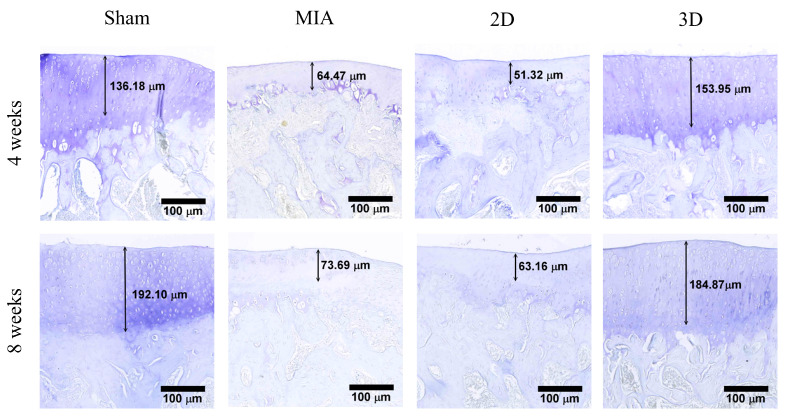
Toluidine blue staining of knee articular cartilage (scale bar = 100 μm).

**Figure 7 ijms-24-07062-f007:**
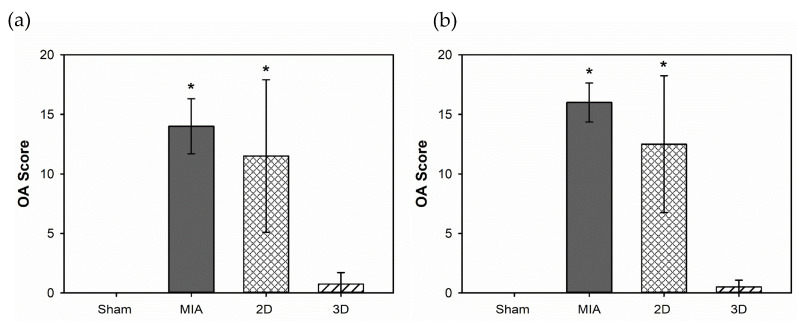
OARSI OA scores evaluation for hADSC spheres treatment on MIA-induced OA rats: (**a**) for 4 weeks; (**b**) for 8 weeks (** p* < 0.05 compared to Sham Group) (n = 4).

**Table 1 ijms-24-07062-t001:** Biochemical and hematological tests for 4 weeks treatment on MIA-induced OA rats (** p* < 0.05 compared to Sham Group).

Item	Unit	Sham	MIA	2D	3D
RBC	10^6^/μL	7.67 ± 0.29	7.73 ± 0.88	7.63 ± 0.51	7.63 ± 0.38
WBC	10^3^/μL	9.08 ± 0.93	9.86 ± 3.39	9.92 ± 1.92	10.18 ± 0.96
PLT	10^3^/μL	866.50 ± 209.56	819.50 ± 454.21	906.75 ± 391.95	911.67 ± 198.61
Monocyte	10^3^/μL	0.46 ± 0.10	0.41 ± 0.14	0.45 ± 0.10	0.49 ± 0.07
Lymphocyte	10^3^/μL	6.53 ± 0.65	6.61 ± 3.03	5.96 ±2.04	7.15 ± 0.47
GOT	U/L	88.33 ± 9.29	110.00 ± 1.41 *	108.50 ± 2.12 *	87.50 ± 7.78
GPT	U/L	28.75 ± 2.22	35.75 ± 5.74	34.00 ± 3.46	30.00 ± 4.40
ALP	U/L	1127.50 ± 428.25	1085.00 ± 268.52	1082.75 ± 224.38	1055.00 ± 344.34
BUN	mg/dL	17.75 ± 2.00	19.80 ± 1.42	19.75 ± 1.42	18.65 ± 2.51
CRE	mg/dL	0.34 ± 0.04	0.28 ± 0.02	0.25 ± 0.03 *	0.28 ± 0.05

RBC—red blood cell; WBC—white blood cell; PLT—platelet; GOT—aspartate aminotransferase; GPT—alanine aminotransferase; ALP—alkaline phosphatase; BUN—blood urea nitrogen; CRE—creatinine.

**Table 2 ijms-24-07062-t002:** Biochemical and hematological tests for 8 weeks treatment on MIA-induced OA rats.

Item	Unit	Sham	MIA	2D	3D
RBC	10^6^/μL	8.11 ± 0.54	7.98 ± 0.59	7.99 ± 0.22	8.32 ± 0.44
WBC	10^3^/μL	6.36 ± 1.50	6.76 ± 0.88	6.65 ± 0.49	6.54 ± 0.75
PLT	10^3^/μL	943.50 ± 67.18	939.33 ± 273.35	937.33 ± 290.18	868.50 ± 62.66
Monocyte	10^3^/μL	0.41 ± 0.09	0.32 ± 0.09	0.37 ± 0.11	0.39 ± 0.05
Lymphocyte	10^3^/μL	4.48 ± 0.86	4.03 ± 0.66	4.17 ± 0.52	4.19 ± 0.46
GOT	U/L	71.25 ± 3.40	67.50 ± 21.02	81.50 ± 19.84	83.50 ± 6.36
GPT	U/L	27.70 ± 2.38	27.75 ± 6.29	27.50 ± 5.92	32.50 ± 1.00
ALP	U/L	890.50 ± 53.03	850.67 ± 154.57	877.00 ± 150.79	928.00 ± 64.63
BUN	mg/dL	18.40 ± 1.14	19.33 ± 3.44	19.27 ± 4.10	18.20 ± 0.84
CRE	mg/dL	0.23 ± 0.02	0.28 ± 0.03	0.27 ± 0.04	0.26 ± 0.03

RBC—red blood cell; WBC—white blood cell; PLT—platelet; GOT—aspartate aminotransferase; GPT—alanine aminotransferase; ALP—alkaline phosphatase; BUN—blood urea nitrogen; CRE—creatinine.

## Data Availability

Not applicable.

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
