# Peer review of "3D-Cultured Adipose-Derived Stem Cell Spheres Using Calcium-Alginate Scaffolds for Osteoarthritis Treatment in a Mono-Iodoacetate-Induced Rat Model"

_ijms, 2023, doi:10.3390/ijms24087062_

Round 1

Reviewer 1 Report

(1)   
Title (Optional): The title could inform readers on the outcome of the study.

(2)   Abstract (Optional): Most findings were very shortly summarized in the abstract section. Biochemical and hematological results, seeding efficiency, SEM analysis and histological findings could have been better mentioned in this section.

(3)   Introduction: Rationale: Thanks to the authors, this section is very clear.

(4)   Introduction: Thanks to the authors, the hypothesis of the study is well written.

(5)   Results: Please indicate p values in Figure 7.

(6)   Discussion: What were the limitations of your study?

Author Response

Response to Reviewer 1 Comments

Dear Reviewer,

    We would like to thank the reviewers for the comments and suggestions for the manuscript. The manuscript has been revised based on your inquires and resubmitted through the journal website. Any changes to the text and figures in the revised manuscript have been marked up using the “Track Changes”. The following is the response to your concerns point-by-point:

Point 1:   Title (Optional): The title could inform readers on the outcome of the study.

Response 1: We appreciate your affirmation, thank you.

Point 2:  Abstract (Optional): Most findings were very shortly summarized in the abstract section. Biochemical and hematological results, seeding efficiency, SEM analysis and histological findings could have been better mentioned in this section.

Response 2: Thank you for your suggestion. However, the word number was 200 words maximum in this section. Therefore, we decide to keep original text in abstract section.

Point 3: Introduction: Rationale: Thanks to the authors, this section is very clear.

Response 3: We appreciate your affirmation.

Point 4: Introduction: Thanks to the authors, the hypothesis of the study is well written.

Response 4: Thank you for your affirmation.

Point 5: Results: Please indicate p values in Figure 7.

Response 5: Thank you for your suggestion, the revision have been made in cption of figure 7. 

Point 6: Discussion: What were the limitations of your study?

Reply: Thank you for your reminder. Indeed, the major limitation was low proliferation rate of hADSC in Ca-Ag scaffold in this study. The limitation was present in Line 277 in revised manuscript.

----------------------------------------------------------------------------------

    Thank you for your valuable comments/suggestions and giving us the opportunity to revise the manuscript to a more readable level. We worked very hard to response your inquires and to revise the manuscript. We hope the manuscript could pass the review to be published in your prestigious journal: International Journal of Molecular Sciences.

Sincerely yours,

Feng-Huei Lin, Ph. D.

Life Distinguished Professor

Institute of Biomedical Engineering, National Taiwan University, Taipei, Taiwan.

E-mail: double@ntu.edu.tw; Tel: 886-2-23912641; Fax: 886-2-23940049

Reviewer 2 Report

In the manuscript by Lin et al, the authors have used human adipose-derived stem cell (hADSC) cultured in 3D system using their in house designed calcium-alginate (Ca-Ag) scaffolds and bioreactor. They injected these cells in monosodium iodoacetate (MIA)-induced osteoarthritis (OA) rat models.   Their results show better out comes with hADSC spheres than the 2D cultured cells using standard OA treatment experiments.  

Here are some of my suggestions that could help to improve the manuscript:

1.       In Figure 1 cartoon, it appears that the hADSC used in this study were isolated from humans directly, however from methods the authors have mentioned that they obtained it from commercial source. So the representation is not correct. Additionally, from the illustration it is also misleading that the 3D spheres were used to treat human OA but in study they used rats. So this cartoon is overall misleading in representing the actual experiments done in this manuscript.

2.        In Figures 2 a-b, the scale values are missing. In 2C what are the arrows showing should either be labeled or indicated in figure legends for example first peak drop is O-H, second is C-H and so on. They have discussed the FT-IR spectrum results in main text but it’s not clear which peak means what since it doesn’t have any other reference/control substance spectrum in the plot.  

3.       In line 134-135 it should be “absorbed liquids” instead of “absorbed of liquid”. More importantly liquid should be defined here whether this is water or saline or any other solution.

4.       In results section 2.2, the authors have mentioned below but none of these data are presented with figure. “Initial cell density was 1 × 106 cells/scaffold and cell seeding efficiency was about 82.67%. The proliferation of hADSCs adhered to the Ca-Ag scaffold was determined after culturing with bioreactor on day 14. The 2.35 fold cell growth from the initial cell density.” Authors should show these data.

5.       Authors have mentioned 4 groups (Sham, MIA, 2D and 3D) in the methods section but they should also mention these groups in main text when they first present their data. From the main text it is not clear what these groups are until one reads the methods section first which often is not the case.

6.       In figure 5, the authors mentioned that they did not see any significant differences between 3D and sham groups. However, at eight weeks the Sham and 3D images look completely different from each other; so I’m not sure on what basis the authors are saying there are no significant differences? Also, which significance test was done to come to these conclusions? In addition, the authors should highlight the chondrocyte and lacunae so the readers can also understand which phenotype authors are referring in these figure that have no significant differences.

7.       For figure 4, how many rats were used for this study? It's a bit surprising to see how just the 3D sphere injections not only improved the OA but also perform better than the control (sham) rats.

8.       In section 2.6, lines 202-203 the authors mentioned that “The results showed no significant difference between the 202 groups.” But from tables it’s not clear which significance test was done to conclude this. From the methods it appear that they have done ANOVA test but authors should mention p-values throughout manuscript wherever they have performed statistical analysis.

9.       Catalog numbers for multiple reagents is missing in the methods sections.

Author Response

Response to Reviewer 2 Comments

Dear Reviewer:

  We would like to thank the reviewers for the comments and suggestions for the manuscript. The manuscript has been revised based on your inquires and resubmitted through the journal website. Any changes to the text and figures in the revised manuscript have been marked up using the “Track Changes”. The following is the response to your concerns point-by-point:

Point 1: In Figure 1 cartoon, it appears that the hADSC used in this study were isolated from humans directly, however from methods the authors have mentioned that they obtained it from commercial source. So the representation is not correct. Additionally, from the illustration it is also misleading that the 3D spheres were used to treat human OA but in study they used rats. So this cartoon is overall misleading in representing the actual experiments done in this manuscript.

Response 1: Thank you for your reminder. The revised cartoon was in Figure 1 of revised manuscript.       

Point 2: In Figures 2 a-b, the scale values are missing. In 2C what are the arrows showing should either be labeled or indicated in figure legends for example first peak drop is O-H, second is C-H and so on. They have discussed the FT-IR spectrum results in main text but it’s not clear which peak means what since it doesn’t have any other reference/control substance spectrum in the plot.  

Response 2: Thank you for your reminder. The revision has been made in figure 2 in revised manuscript.

Point 3: In line 134-135 it should be “absorbed liquids” instead of “absorbed of liquid”. More importantly liquid should be defined here whether this is water or saline or any other solution.

Response 3: The you for your suggestion, we change the “absorbed PBS” instead of “absorbed of liquid” in the revised manuscript.

Point 4:  In results section 2.2, the authors have mentioned below but none of these data are presented with figure. “Initial cell density was 1 × 106 cells/scaffold and cell seeding efficiency was about 82.67%. The proliferation of hADSCs adhered to the Ca-Ag scaffold was determined after culturing with bioreactor on day 14. The 2.35 fold cell growth from the initial cell density.” Authors should show these data.

Response 4: Thank for your suggestion, the diagram figures of cell proliferation was shown in supplement data.

Point 5:  Authors have mentioned 4 groups (Sham, MIA, 2D and 3D) in the methods section but they should also mention these groups in main text when they first present their data. From the main text it is not clear what these groups are until one reads the methods section first which often is not the case.

Response 5: Thank you for your reminder. The group definition was moved from method section to results section in the revised manuscript.

Point 6: In figure 5, the authors mentioned that they did not see any significant differences between 3D and sham groups. However, at eight weeks the Sham and 3D images look completely different from each other; so I’m not sure on what basis the authors are saying there are no significant differences? Also, which significance test was done to come to these conclusions? In addition, the authors should highlight the chondrocyte and lacunae so the readers can also understand which phenotype authors are referring in these figure that have no significant differences.

Response 6: Thank you for your reminder. We should use the “significant” more carefully. We addressed the section 2.4 and figure 5 in revised manuscript, and illustrated the chondrocyte and empty lacunae in part of Supplementary Materials.

Point 7:  For figure 4, how many rats were used for this study? It's a bit surprising to see how just the 3D sphere injections not only improved the OA but also perform better than the control (sham) rats.

Response 7: Thank you for your reminder. For each group, we performed 4 rats in this animal study. Figure 4 showed 3D group could improve the MIA-induced OA. However, there is no significant difference between 3D and sham group. We estimated the phenomenon is due to the individual differences in behavior test.   

Point 8: In section 2.6, lines 202-203 the authors mentioned that “The results showed no significant difference between the 202 groups.” But from tables it’s not clear which significance test was done to conclude this. From the methods it appear that they have done ANOVA test but authors should mention p-values throughout manuscript wherever they have performed statistical analysis.

Response 8: Thank you for your suggestion. We revised the section 2.6 and Tables, and mentioned the p value in revised manuscript.

Point 9: Catalog numbers for multiple reagents is missing in the methods sections.

Response 9: Thank you for your reminder. All the catalog numbers of reagents were added in methods section of revised manuscript.

- - - - - - - - - - - - - - - - - - - - - - - - - - - - - - - - - - - - - - - - - - - - - - - - - - - - - - - - - - - - - - - - - - - -

  Thank you for your valuable comments/suggestions and giving us the opportunity to revise the manuscript to a more readable level. We worked very hard to response your inquires and to revise the manuscript. We hope the manuscript could pass the review to be published in your prestigious journal: International Journal of Molecular Sciences.

Sincerely yours,

Feng-Huei Lin, Ph. D.

Life Distinguished Professor

Institute of Biomedical Engineering, National Taiwan University, Taipei, Taiwan.

E-mail: double@ntu.edu.tw; Tel: 886-2-23912641; Fax: 886-2-23940049
